# Colorimetric Detection of Nitrosamines in Human Serum Albumin Using Cysteine-Capped Gold Nanoparticles

**DOI:** 10.3390/s25175505

**Published:** 2025-09-04

**Authors:** Sayo O. Fakayode, David K. Bwambok, Souvik Banerjee, Prateek Rai, Ronald Okoth, Corinne Kuiters, Ufuoma Benjamin

**Affiliations:** 1Department of Chemistry and Physics, College of Arts and Sciences, University of North Carolina at Pembroke, 1 University Drive, Pembroke, NC 28372, USA; 2Department of Chemistry, Ball State University, Muncie, IN 47306, USA; david.bwambok@bsu.edu (D.K.B.); ufuoma.benjamin@bsu.edu (U.B.); 3Molecular Biosciences, Middle Tennessee State University, Murfreesboro, TN 37132, USA; souvik.banerjee@mtsu.edu (S.B.); pr3x@mtmail.mtsu.edu (P.R.); 4Department of Chemistry, Middle Tennessee State University, Murfreesboro, TN 37132, USA; 5Department of Chemistry, Physics and Astronomy, Georgia College and State University, Milledgeville, GA 31061, USA; ronald.okoth@gcsu.edu (R.O.); corinne.kuiters@bobcats.gcsu.edu (C.K.)

**Keywords:** nitrosamines, N-nitroso-diethylamine, cysteine–gold-nanoparticles, colorimetry, serum albumin, molecular docking, molecular dynamic simulation

## Abstract

Nitrosamines, including N-nitroso diethylamine (NDEA) have emerged as pharmaceutical impurities and carcinogenic environmental contaminants of grave public health safety concerns. This study reports on the preparation and first use of cysteine–gold nanoparticles (CysAuNPs) for colorimetric detection of NDEA in human serum albumin (HSA) under physiological conditions. Molecular docking (MD) and molecular dynamic simulation (MDS) were performed to probe the interaction between NDEA and serum albumin. UV–visible absorption and fluorescence spectroscopy, dynamic light scattering (DLS), and transmission electron microscopy (TEM) imaging were used to characterize the synthesized CysAuNPs. These CysAuNPs show a UV–visible absorbance wavelength maxima (λ_max_) at 377 nm and emission λ_max_ at 623 nm. Results from DLS measurement revealed the CysAuNPs’ uniform size distribution and high polydispersity index of 0.8. Microscopic imaging using TEM showed that CysAuNPs have spherical to nanoplate-like morphology. The addition of NDEA to HSA in the presence of CysAuNPs resulted in a remarkable increase in the absorbance of human serum albumin. The interaction of NDEA–CysAuNPs–HSA is plausibly facilitated by hydrogen bonding, sulfur linkages, or by Cys–NDEA-induced electrostatic and van der Waal interactions. These are due to the disruption of the disulfide bond linkage in Cys–Cys upon the addition of NDEA, causing the unfolding of the serum albumin and the dispersion of CysAuNPs. The combined use of molecular dynamic simulation and colorimetric experiment provided complementary data that allows robust analysis of NDEA in serum samples. In addition, the low cost of the UV–visible spectrophotometer and the easy preparation and optical sensitivity of CysAuNPs sensors are desirable, allowing the low detection limit of the CysAuNPs sensors, which are capable of detecting as little as 0.35 µM NDEA in serum albumin samples, making the protocol an attractive sensor for rapid detection of nitrosamines in biological samples.

## 1. Introduction

The pharmaceutical industry is committed to the current Good Manufacturing Practices (cGMP) [1,2,3] and other best practice guidelines to ensure pharmaceutical drug quality, safety, and integrity of medications [4,5,6,7,8]. However, drug impurities such as nitrosamines cannot be eliminated during organic chemical synthesis or drug manufacturing processes, leading to drug contamination and adverse side effects [9,10]. Nitrosamines are widespread pharmaceutical impurity contaminants, typically formed as a byproduct of organic reactions or potential nitrosating agents such as nitrites that can react with amines in pharmaceutical excipients at elevated temperatures or under an acidic medium [11,12,13,14,15,16,17,18,19,20,21]. Concerns about nitrosamine pharmaceutical drug contamination arise because nitrosamines are reactive organic chemicals that are carcinogenic, mutagenic, teratogenic, and genotoxic [6,22,23,24,25] and which have resulted in the recall of several medications, including valsartan, irbesartan, losartan, metformin, ranitidine, and nizatidine [7,13,26,27,28]. Nitrosamines have also been detected in tobacco products [29,30], drinking water [31,32], waste and river water samples [33,34], and processed meat and poultry products [35,36,37]. Nitrosamines are, therefore, toxic contaminants of grave environmental and public health safety concerns.

High performance liquid chromatography-mass spectrometry (HPLC-MS) [38,39] and gas chromatography-mass spectrometry (GC-MS) [40,41,42,43,44,45,46] are the most widely used analytical techniques for routine nitrosamine analyses. The practical applications of electrochemical sensors for nitrosamine detection in various samples have also been reported [47]. HPLC-MS and GC-MS methods have proved efficient and accurate for nitrosamine analyses. However, in addition to high cost, longer analysis time, and skilled personnel training, HPLC-MS and GC-MS instruments are not portable, which preclude their use for fast, on-site, point-of-care, and field analysis of nitrosamines. Using a simple, rapid, low-cost colorimetric assay may address some of the current challenges of HPLC-MS and GC-MS methods for nitrosamine sample analysis. Beard et al. [48] recently demonstrated the use of colorimetric assay for sensitive and selective NDEA detection in aqueous solutions via photo nitrosation using a naphthol sulfonate indicator. The potential use of chemiluminescence detection of nitrosamines in wastewater samples was also reported [49]. Colorimetric chemical detection often capitalizes on the optical properties of the analytes or reagent or relies on the changes (enhancement or decrease) in the optical properties of the analytes–reagent interaction. The ease of preparation, unique optical and magnetic properties, biological affinity, and molecular recognition capacities of gold nanoparticles (AuNPs) make AuNPs highly desirable for colorimetric analyte detection in environmental samples, bioanalytical and biomedical studies, and clinical diagnoses [50]. In addition, AuNPs are tunable and can easily be modified or tailored with various moieties including spin-coated poly (vinyl alcohol)/chitosan [51,52], citrate [53], dopamine [54], and PEGylated [55] to promote AuNP’s stability in solutions, to enhance AuNP’s optical properties, and to achieve greater analytical sensitivity and selectivity. Nonetheless, cysteine-capped AuNPs (CysAuNPs) remain one of the most widely used sensors for colorimetric chemical sensing and analyte detection in various matrices. To our knowledge, the use of CysAuNPs as colorimetric sensors for detecting nitrosamines in biological systems such as human serum albumin (HSA) samples has not yet been explored. The unique chemical structure [56] allows HSA to bind, transport, and distribute various metabolites, waste materials, and toxic materials to various targets in human serum albumin proteins which can potentially bind, transport, and distribute NDEA to various targets in the human body [57,58,59,60,61,62]. The interaction of NDEA with serum albumin can potentially impact the molecular structure, physiology, and functionality of serum albumin with adverse effects. This study reports the synthesis and first use of CysAuNPs for colorimetric detection of NDEA in human serum albumin (HSA), the most abundant serum protein in the cardiovascular system and a transporter of drugs and metabolites to various targets in the human body, at physiological conditions. Human serum albumin was selected because of its physiological role in humans. As a drug and a metabolite transporter in the human body, serum aluminum can potentially bind with NDEA, with deleterious health implications for humans. Colorimetric analysis is low-cost and involves simple sample preparation and direct sample measurement. Molecular docking (MD) and molecular dynamic simulation (MDS) were performed to further probe and gain insight into the molecular interaction between NDEA and serum albumin. CysAuNPs sensors were selected for NDEA detection in this study because of the attractive optical properties of CysAuNPs that allow sensitive detection of analytes at low concentrations and the stability of CysAuNPs in solution. The combined use of molecular dynamic simulation and colorimetric experiment provided complementary data that allow robust analysis of NDEA in serum samples. In addition, the low cost of the UV–visible spectrophotometer, easy preparation, and optical sensitivity of CysAuNPs sensors in this study are desirable, making the reported protocol an attractive sensor for rapid detection of nitrosamines in biological samples.

## 2. Material and Methods

*Chemicals and Reagents:* Analytical grade reagents including N-nitroso diethylamine (NDEA), human serum albumin (HSA) L-cysteine, gold tetrachloride (HAuCl_4_) and 3-(N-morpholino) propanesulfonic acid (MOPs) were purchased from Sigma-Aldrich (Saint Louis, MO, USA) and used as received. Formvar copper grids were purchased from TedPella (Redding, CA, USA).

*Synthesis of CysAuNPs*: The cysteine–gold nanoparticles (CysAuNPs) were synthesized by modification of a reported procedure [63]. A stock solution of cysteine (100 mM) was prepared in water. Separately, a stock solution of 20 mM of HAuCl_4_ was prepared in water. A volume of 5 mL of an aqueous solution of cysteine (100 mM) and 5 mL of aqueous solution of HAuCl_4_ (20 mM) were added into a clean round-bottomed flask equipped with a stirring bar. The mixture was stirred to obtain a homogenous solution at room temperature for 72 h. The solution color changed slowly from colorless to milky white. The product was purified by centrifugation at 10,000 rpm for 6 min, followed by washing with deionized water, then centrifugation. The supernatant was discarded to obtain the CysAuNPs residue. The resultant CysAuNPs precipitate was re-dispersed in deionized water (2 mL) and ultrasonicated for 30 min to obtain uniform CysAuNPs. The synthesis was scaled up to obtain a liter of the CysAuNPs dispersed in water, which was used for all the measurements reported in the study.

*Preparation of NDEA, HSA samples:* All studies were conducted at physiological temperature (37 °C) and pH 7.4. The human serum albumin solution was prepared in 30 mM MOPS buffer at pH 7.4 and refrigerated for 48 h before use to ensure the stability of the solution. Working HSA concentration range samples were prepared by serial dilution of the serum stock solution in 30 mM MOPS buffer, pH 7.4. HSA concentrations were accurately determined from UV–visible absorption spectra using Beer’s law. NDEA stock solution was prepared using spectroscopic-grade methanol (purity > 99.999%) for solubility consideration. Two sets of samples were prepared for the study. The first sample set contained a varying NDEA concentration in a fixed 9.64 µM HSA sample. In contrast, the second sample set contained the varying NDEA concentrations in a fixed 9.64 µM HSA solution. Each sample received 25 µL of CysAuNPs solution. A control sample was prepared by pipetting 25 µL of CysAuNPs solution into fixed 9.64 µM HSA concentration but without any NDEA analyte.

*Sample spectral measurements:* All UV–visible absorption spectra measurements were performed with a UV–visible spectrophotometer (Shimadzu, Kyoto, Japan, UV-2600i) with a temperature-controlled device accessory. The emission data were collected using a spectrofluorometer (Jasco FP-8550, Tokyo, Japan) with a temperature-controlled device. A 1 cm all-clear glass cuvette was used for fluorescence measurements.

*Sample preparations for characterization of the CysAuNPs:* The CysAuNPs were diluted in water and sonicated for 2 min to disperse the nanoparticles before characterization by UV–visible absorption, fluorescence, DLS, and TEM. The dilute nanoparticle suspensions were placed in a 1 cm quartz cuvette for UV–visible and DLS measurements. The nanoparticle suspension was transferred with a syringe into a Zeta potential capillary cell to measure surface charge using a DLS instrument. For TEM measurements, a diluted sample of the CysAuNPs was drop-cast onto a Formvar copper grid and allowed to dry in air at room temperature before imaging. The size and zeta potential of the nanoparticles were measured using a Zetasizer Nano ZS dynamic light scattering (DLS) instrument (Malvern, Malvern, UK). Transmission electron microscopy (TEM) imaging for the determination of size and morphology of the nanoparticles was performed on a JEOL TEM instrument (JEOL, Tokyo, Japan) at an operating voltage of 120 kV.

*Molecular docking:* The three-dimensional (3D) structure of NDEA was constructed and optimized through MM2 energy minimization in Avogadro modeling software (version 1.2.0) [64]. The 3D structure of the target protein, Human Serum Albumin (PDB ID: 1AO6), was sourced from the RCSB Protein Data Bank (PDB). Energy minimization of the protein structure was further refined using the GROMOS96 force field in Swiss PDB-Viewer 4.1 [65], which effectively resolved bad contacts and corrected unfavorable torsion angles. The binding pockets of HSA for NDEA were predicted using the CB-Dock2 web server, which employs cavity detection through blind docking [66]. CB-Dock2 performed docking of NDEA at the predicted binding sites and estimated the binding affinity based on AutoDock Vina’s binding energy [67]. The protein–ligand complexes docked at the top two predicted binding sites were subjected to molecular dynamics simulations (MDS) to evaluate their affinity during a 200 ns production run.

*Molecular dynamics simulations and binding free energy calculations:* Protein–ligand complexes docked at the top two predicted binding sites underwent (MDS) to assess their binding affinity during a 200 ns production run. The simulations were carried out in GROMACS (version 2024.2) [68] using the CHARMM36 force field [69] for proteins and CGenFF for ligands [70]. The system was prepared in a dodecahedron box with periodic boundaries, solvated with TIP3P water, and neutralized with Na^+^ and Cl^−^ ions. After parameterization, energy minimization was performed with 5000 steps of steepest descent and 10,000 steps of conjugate gradient to resolve unfavorable interactions. Annealing under NVT conditions for 50 ps raised the temperature to 300 K, followed by 1 ns of density equilibration under NPT conditions (300 K, 1 atm) using a Langevin thermostat (2 ps collision frequency) and Berendsen barostat (1 ps relaxation). Long-range van der Waals and electrostatic interactions were managed using the particle mesh Ewald method with a 1.2 nm cutoff. The 200 ns production run was complemented with additional replicas initiated from varied random velocities for validation.

Binding free energy calculations were conducted on the converged trajectories obtained through the replicates of the MDS using the MM-GBSA (Molecular Mechanics Generalized Born Surface Area) approach implemented in the gmx_MMPBSA tool [71]. These computations accounted for van der Waals interactions from molecular mechanics, electrostatic energy, and the solvation-free energy contributions, including both the electrostatic component (via the Generalized Born equation) and the nonpolar component (using an empirical model). Water molecules and ions were excluded from the system prior to the MM-GBSA analysis.

## 3. Results and Discussion

### 3.1. Characterization of Synthesized CysAuNPs

*Size, morphology, and surface charge of the synthesized CysAuNPs:* The average hydrodynamic diameter of the CysAuNPs determined by DLS measurements was 282 nm. The distribution of particle sizes resulted in a polydispersity index (PDI) of 0.1. The CysAuNPs have a high negative surface charge of −33.7 (±4.70) mV, suggesting that these CysAuNPs are stable when dispersed in water (Figure 1). This is consistent with reported negative zeta potential for CysAuNPs and other amino acid-AuNPs in solutions [72,73]. In addition, cysteine is a reducing agent and stabilizes the AuNPs [74,75].

Knowledge of the surface charge can provide insights into the interaction mechanism of the CysAuNPs used to detect nitrosamines in this study. The size and morphology of the nanoparticles were determined by TEM measurement after drop-casting the CysAuNPs suspension on the Fomvar copper grid and allowing it to dry in air at room temperature. The CysAuNPs displayed spherical to nanoplate-like morphology (Figure 1). The shape of these nanoparticles is similar to that of previously reported CysAuNps [63].

The UV–visible absorption spectra measurements of CysAuNPs diluted in water showed an absorbance maximum at a wavelength of 377 nm. There is also another peak at 241 nm (Figure 2). These peaks are ascribed to metal-centered or ligand–metal charge transfer transitions [76]. The fluorescence emission of CysAuNPs diluted in water was obtained at 623 nm following excitation at 377 nm (Figure 2). The Stokes shift was 246 nm, which is close to the 250 nm reported for thiol–gold complexes resulting from metal–ligand (gold–thiol) charge transfer and metal–metal interactions [76]. In addition, aggregation of CysAuNPs can be facilitated by Cys–Cys-induced electrostatics interaction [77]. Taken together, DLS, TEM, UV–visible, and fluorescence spectroscopy characterization results confirm the successful synthesis of CysAuNPs. In this study, the prepared CysAuNPs were used for colorimetric detection of nitrosamines.

### 3.2. Colorimetric Detection of NDEA in HSA Samples

Figure 3A shows the UV–visible absorption spectra of solutions that contained a fixed HSA concentration (9.64 µM) and varying NDEA concentrations ranging from 2.46 µM to 8.61 µM. HSA shows its characteristic UV–visible absorption, with λ_max_ at 280 nm due to tryptophan residue [78]. NDEA has weak UV–visible absorption. The interaction of NDEA with HSA has minimal impact on the HSA UV–visible absorption. However, the addition of NDEA and CysAuNPs into serum albumin resulted in a cloudy white solution, which is more conspicuous at higher NDEA concentrations, suggesting serum albumin protein agglomeration.

Figure 3B shows the UV–visible absorption of 9.64 µM HSA and CysAuNPs–HSA samples. Adding 25 µL of CysAuNPs to 9.64 µM HSA solution significantly increased HSA UV–visible spectra absorption, with a slight blue spectrum shift toward a shorter wavelength. The shift in absorption spectrum of CysAuNPs–HSA suggest changes in the HSA micromolecular environment because of its interaction with CysAuNPs. In addition, CysAuNPs–HSA shows an intense absorption at a long-wavelength range (314 nm to 500 nm range) with a λ_max_ at 380 nm that was not present in absorption spectrum of serum albumin. The results of our absorption spectra of CysAuNPs–HSA were consistent with prior studies that have reported a similar blue shift and an increase in the absorption of CysAuNPs–HSA [79]. Figure 3C shows the UV–visible absorption spectra of solutions containing a fixed 9.64 µM HSA and varying NDEA concentrations ranging from 2.46 µM to 8.61 µM, like the solutions in Figure 3A. However, each solution in Figure 3C also received 25 µL of CysAuNPs. The NDEA–CysAuNPs–HSA solutions have a remarkably higher UV–visible absorption than the corresponding CysAuNPs–HSA solutions.

Figure 4 shows an overlay of UV–visible absorption spectra of HSA, HSA–NDEA, CysAuPNs–HSA, and NDEA–CysAuPNs–HSA solutions at different NDEA concentrations. Figure 5A is the net absorption of NDEA–CysAuPNs–HSA samples with a λ_max_ at 378 nm. The net UV–visible absorption spectra were calculated by subtracting the absorbance value of the CysAuNPs–HSA solution from the absorbance value of the NDEA–CysAuNPs–HSA solution for each NDEA concentration. The net resultant UV–visible absorption was calculated to evaluate the impact of NDEA concentration on NDEA–CysAuNPs–HSA absorption spectra. Figure 5B depicts net absorbance values versus NDEA concentration in NDEA–CysAuNPs–HSA. The net absorbance value plot shows a double calibration curve of two interesting NDEA concentration regions. In the first region (2.46 µM to 4.92 µM NDEA concentrations), the net absorbance of the solution decreases progressively with NDEA concentrations, reaching the minimum value at 4.92 µM NDEA (y = −0.0415x + 0.2433; R^2^ = 0.8643). The observed double calibration curve in Figure 5B indicates that a single linear model cannot represent the data due to multiple reasons, including changes in instrumental sensitivity and response to NDEA concentration. A similar double calibration curve has been reported in the literature [80]. Interestingly, the net minimum absorbance was obtained when the molar concentration ratio of NDEA and HSA in NDEA–CysAuNPs–HSA is 2:1. A decrease in absorbance at 4.92 µM NDEA concentration is likely due to changes in dynamics and equilibrium due to NDEA interaction with serum albumin. In the second region (4.92 µM to 8.61 µM NDEA), the net absorbance increases linearly with NDEA concentrations in NDEA–CysAuNPs–HSA solution (y = 0.0179x − 0.036; R^2^ = 0.9938) (Figure 5B). Depending on the NDEA concentrations, two possible calibration curves can be constructed for the determination of unknown NDEA samples. Using the first calibration curve, the obtained figures-of-merit highlighted by a high correlation, the calculated low limit of detection (LOD) of 0.210 µM, and the limit of quantitation (LOQ) of 0.70 µM show the sensitivity of CysAuNPs for NDEA detection in serum albumin. The LOD was calculated as $\frac{3s}{m},$ where *s* is the standard deviation of blank triplicate analyses and *m* is the slope of the calibration curve regression equation. The LOQ was calculated as $\frac{10s}{m},$ where *s* is the standard deviation of blank triplicate analyses and *m* is the slope of the calibration curve regression equation. The use of the second calibration curve resulted in a LOD of 0.49 µM and a LOQ of 1.62 µM for NDEA detection in serum albumin. Either calibration curve can be used for NDEA detections depending on the NDEA concentration range. Our observed LOD is comparable with the reported limit of detection of 0.66 pm colorimetric detection of N-nitrosamines in aqueous water [48] and a limit of detection LOD ranging between 0.06 and 0.5 µM for chemiluminescence detection of N-nitrosamines in water samples [49]. A fluorescence study will lower the detection limit and provide better selectivity.

### 3.3. Job Plot

A Job plot of the net absorbance versus mole fraction of [NDEA] in [NDEA–CysAuNPs–HSA] was plotted to gain further insight into the NDEA–CysAuNPs–HSA interaction in solution (Figure 4C). The Job plot indicated a 1:2 mole ratio of NDEA to HSA in NDEA–CysAuNPs–HSA.

### 3.4. Proposed NDEA–CysAuNPs–HSA Interaction Mechanism

As illustrated in Figure 6, the aggregation of CysAuNPs is driven by electrostatic interactions between zwitterionic cysteine moieties [81,82]. This same zwitterionic interaction has also been proposed to explain the aggregation mechanism of cysteine-capped gold nanorods [83]. When CysAuNPs are introduced to HSA, the gold nanoparticles can bind to the protein through formation of Au–sulfur bonds either at the free Cys-34 or at one of the four disulfide-linked Cys–Cys pairs [84]. Previous studies have shown that this binding can induce conformational changes or partial unfolding of the HSA protein [79,84]. In our studies, we observed that these conformational changes or partial unfolding events are more pronounced in the presence of NDEA, resulting in a measurable colorimetric response.

We hypothesize that the Cys on the surface of the AuNPs aggregate experience reduced electrostatic interactions between zwitterionic cysteine moieties. It is plausible that NDEA may dislodge some Cys–AuNPs on the surface of the aggregate through electrostatic interactions between NDEA and cysteine. Once dislodged, the ‘free’ NDEA-CysAuNPs complexes are less sterically hindered and can bind more effectively to HSA. Our data suggests that the dimerization of HSA (see Job’s plot in Figure 5C) upon binding to these ‘free’ CysAuNPs–NDEA complexes initiates a cascade of events that culminate in the observed colorimetric response. We propose that this response is triggered by an initial conformational change in HSA upon binding. This initial conformational change likely facilitates the penetration of NDEA molecules into the protein’s interior domains, disrupting its native H-bonding networks and electrostatic interactions. The result is a higher-order unfolding of the HSA protein. We attribute the observed colorimetric response of CysAuNPs–HSA conjugate to this extensive unfolding, which is driven by high concentrations of NDEA.

### 3.5. MD and MDS Results

*Prediction of the binding site of NDEA within the HSA protein.* To identify the potential binding site of NDEA within the HSA protein, blind docking was initially performed using the Cb-Dock2 webserver. Subsequently, the compound was docked at all five predicted binding sites using AutoDock Vina. The top two binding sites for NDEA on HSA both yielded a Vina score of −4.3 kcal/mol (Figure 7). To validate the docking predictions, 200 ns MD simulations (MDS) were conducted.

Analysis of the RMSD values revealed that the first binding site demonstrated greater stability, with average RMSDs of 2.8 Å for the protein backbone, 0.9 Å for the ligand, and 3.1 Å for the complex (Figure 8A). At site 1, NDEA was surrounded by residues TYR150, LEU238, ARG257, SER287, ILE290, and ALA291. Notably, NDEA formed bifurcated hydrogen bonds with TYR150 and ARG257 through its nitroso oxygen, contributing to its stable binding. In contrast, the second binding site exhibited average RMSDs of 3.0 Å, 0.8 Å, and 10.0 Å for the protein backbone, ligand, and complex, respectively (Figure 8B). Site 2, on the other hand, included residues ASP108, HIS146, PRO147, and ARG197. Unlike site 1, NDEA did not form any hydrogen bonds at this site, which likely contributed to the higher RMSD observed in the MD simulations and the less stable binding free energy. The significantly higher RMSD for the complex at the second site indicates that the ligand moved considerably from its initial position. The movement of NDEA between binding sites may be driven by weaker and less stable interactions at site 2. As noted earlier, no hydrogen bonding interactions were observed at this site. Additionally, the MM-GBSA-derived binding free energy supports this observation, indicating lower stability at site 2. This stability at the first site was further supported by MM-GBSA-derived binding free energy. The average ΔG at the first site was −14.5 kcal/mol, compared to −12.2 kcal/mol at the second site (Table 1). MM-GBSA calculations do not provide absolute binding free energy, and the binding free energy obtained from the two sites is comparable. Therefore, there are possibilities that NDEA is binding to both the sites simultaneously. Collectively, these results suggest that the first binding site is the most likely binding site for NDEA within the HSA protein. The Job plot indicating a 2:1 NDEA:HSA stoichiometry provides experimental evidence that more than one NDEA molecule can associate with the HSA protein. This observation aligns with our molecular docking and MD simulation results, which revealed that NDEA can bind at two distinct sites on HSA with comparable binding free energies. These findings suggest that HSA can simultaneously accommodate multiple NDEA molecules at different binding regions, which explains the 1:2 stoichiometry detected experimentally. Thus, the congruence of the Job plot data with the computational binding site predictions strengthens the mechanistic interpretation of NDEA–HSA interactions and their role in driving the optical response of the CysAuNP-based sensor.

## 4. Conclusions

In summary, CysAuNPs were synthesized and characterized for colorimetric detection of NDEA nitrosamine in serum albumin. The Job plot, molecular docking, and molecular dynamics simulation results suggest a 1:2 mole ratio complexation between NDEA and HSA. The interactions of NDEA–CysAuNPs–HSA are facilitated by Cys–NDEA-induced electrostatic interaction due to Cys–Cys interaction disruption upon the addition of NDEA. However, other forces, including hydrogen bonding, sulfur linkages, π-electron, and hydrophobic interaction, may promote the binding of NDEA with serum albumin in the presence of CysAuNPs. The low-cost and improved miniaturized UV–visible spectrometer technology, easy preparation of CysAuNPs, and the capability of CysAuNPs sensors to detect as little as 0.35 µM NDEA in serum albumin samples make the reported protocol in this study attractive for rapid and sensitive detection of nitrosamines in serum albumin. Future studies include using CysAuNPs to detect more nitrosamines in real biological samples including human serum, urine samples, environmental, and pharmaceutical samples. Future studies also include the determination of the binding affinity and the use of circular dichroism spectroscopy to probe the molecular interaction of NDEA binding with serum albumin, which may provide additional insight into the binding mechanism and structural and molecular changes of serum albumin due to interaction with NDEA. These are ongoing research studies and will be communicated in future manuscripts.

## Figures and Tables

**Figure 1 sensors-25-05505-f001:**
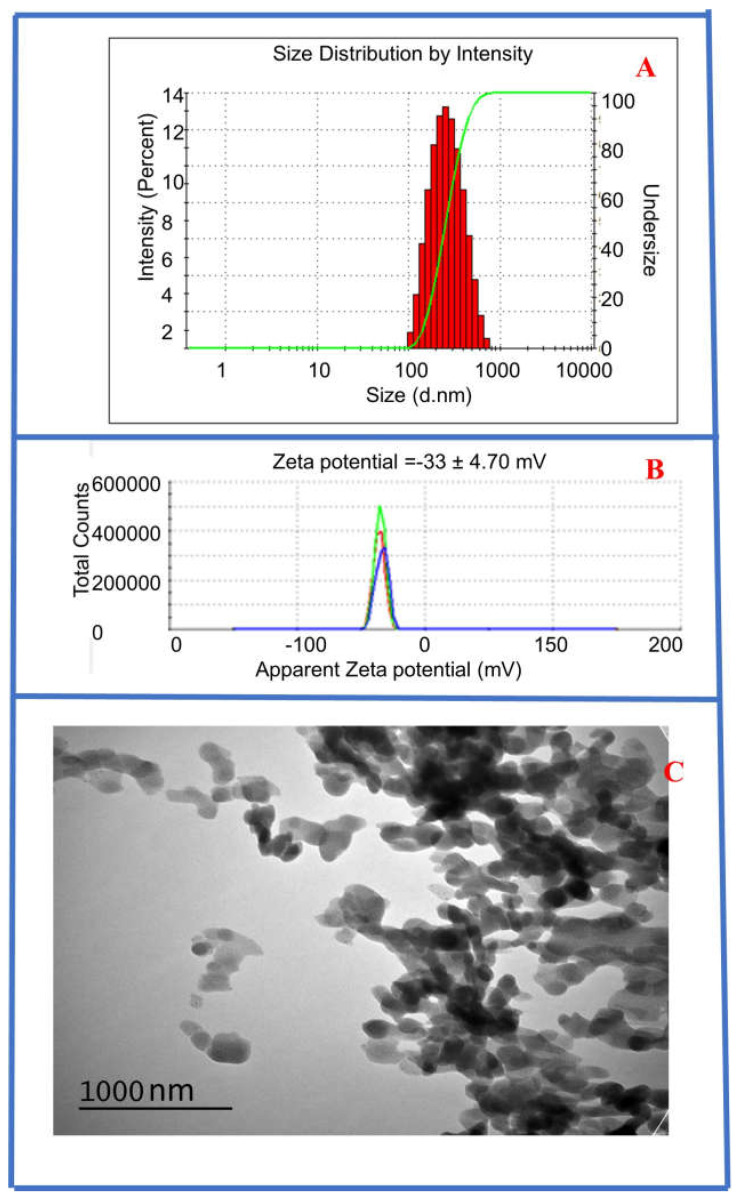
(**A**) DLS measurement of CysAuNPs showing an average hydrodynamic diameter of 282 nm. (**B**) Zeta potential of CysAuNPs showing surface charge of −33.7 mV. (**C**) TEM of CysAuNPs (at 120 kV, 200 K magnification) showing spherical, nanoplate-like morphology of CysAuNPs.

**Figure 2 sensors-25-05505-f002:**
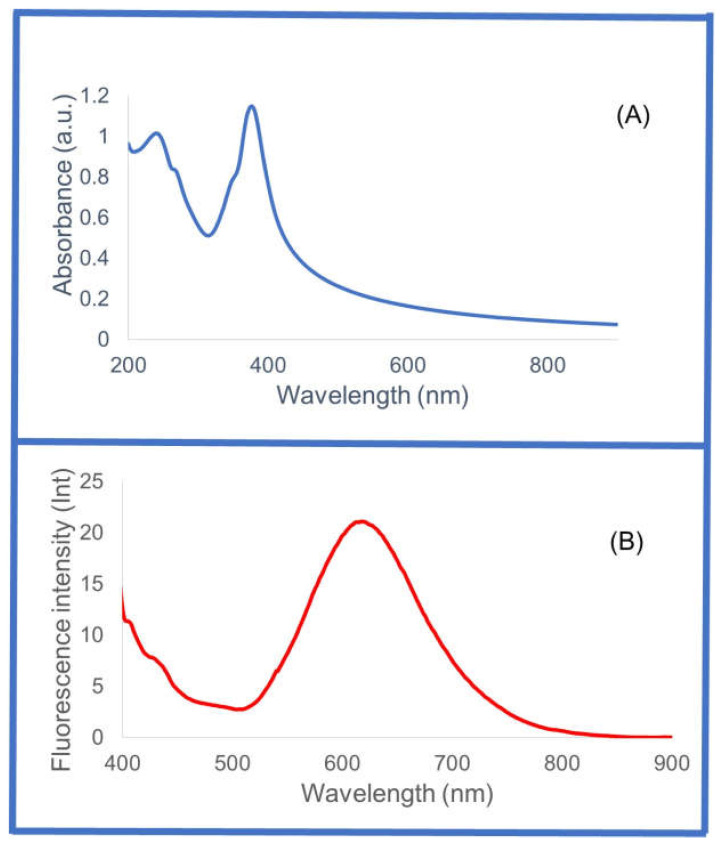
(**A**) UV–vis absorbance of synthesized cysteine–gold nanoparticles diluted in water. The CysAuNPs have maximum absorbance at 377 nm and another peak at 241 nm. (**B**) Fluorescence emission of CysAuNPs is at 623 nm (using 377 nm as the excitation wavelength).

**Figure 3 sensors-25-05505-f003:**
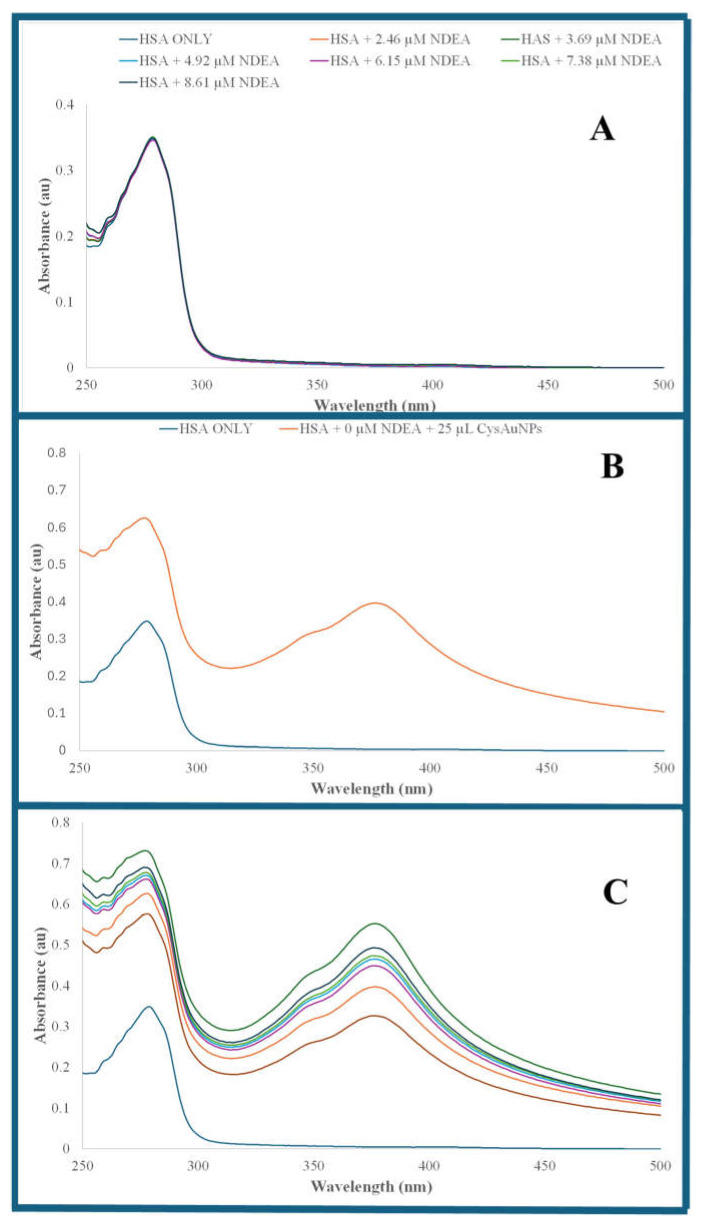
UV–visible absorption spectra of (**A**) solution containing fixed 9.64 µM of HSA and varying NDEA concentrations ranging from 2.46 µM to 8.61 µM NDEA; (**B**) 9.64 M HSA and 9.64 µM HSA+ L CysAuNPs; (**C**) 9.64 M HSA, 9.64 µM HSA+ L CysAuNPs and varying NDEA concentrations ranging from 2.46 µM to 8.61 µM NDEA like in Figure 3A.

**Figure 4 sensors-25-05505-f004:**
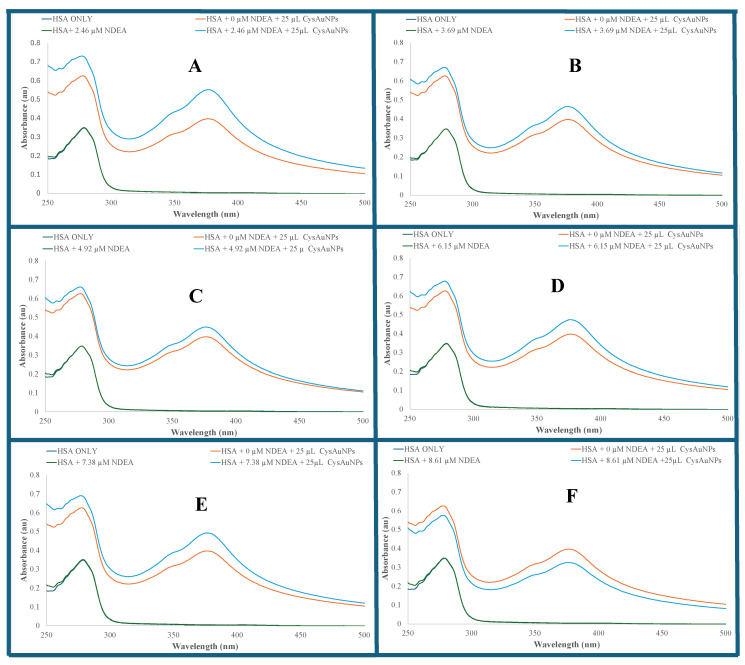
Overlay of UV–visible spectra of 9.64 µM HSA, HSA–NDEA, CysAuNPs–HSA and NDEA–CysAuNPs–HSA solutions at different NDEA concentrations. Each solution contained the same 9.64 µM HSA concentration. The different NDEA concentrations used are: (**A**) 2.46 µM, (**B**) 3.69 µM, (**C**) 4.92 µM, (**D**) 6.15 µM, (**E**) 7.38 µM, and (**F**) 8.61 µM.

**Figure 5 sensors-25-05505-f005:**
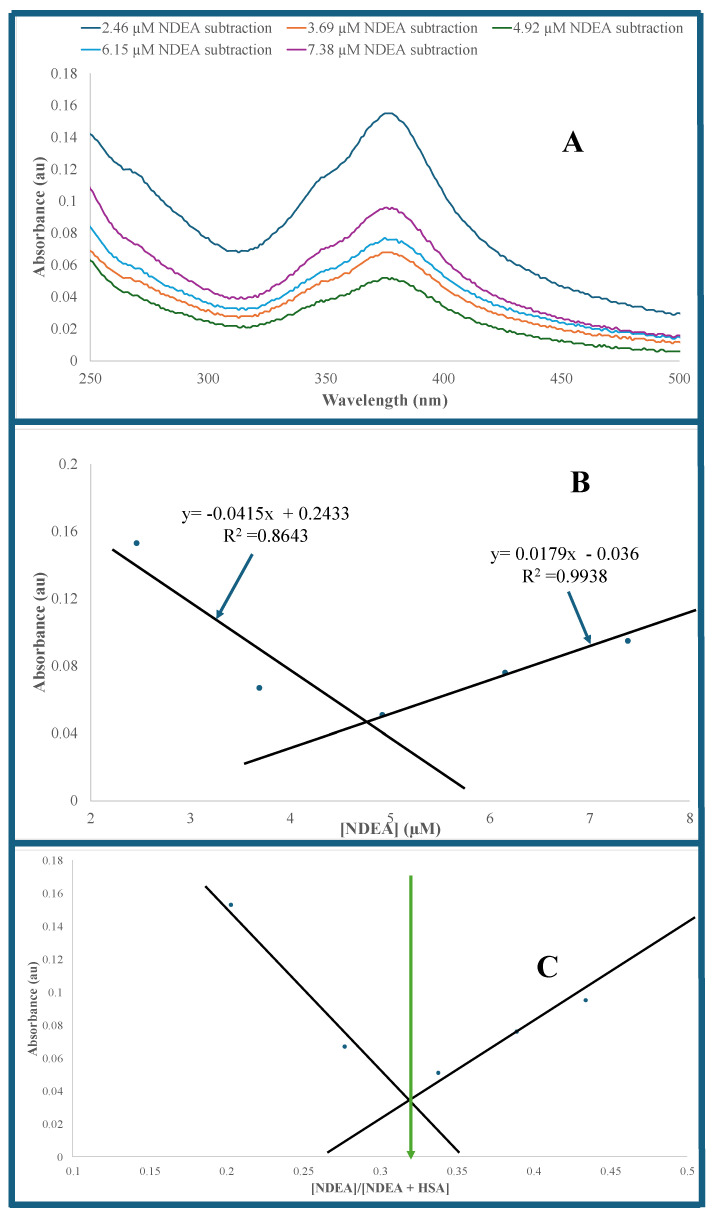
(**A**). Overlay of the UV-visible spectra after subtraction of NDEA-CysAuNPs-HSA and CysAuNPs-HSA. (**B**). The net absorbance versus NDEA concentration. (**C**). Job plot of absorbance versus mole fraction of NDEA in solution.

**Figure 6 sensors-25-05505-f006:**
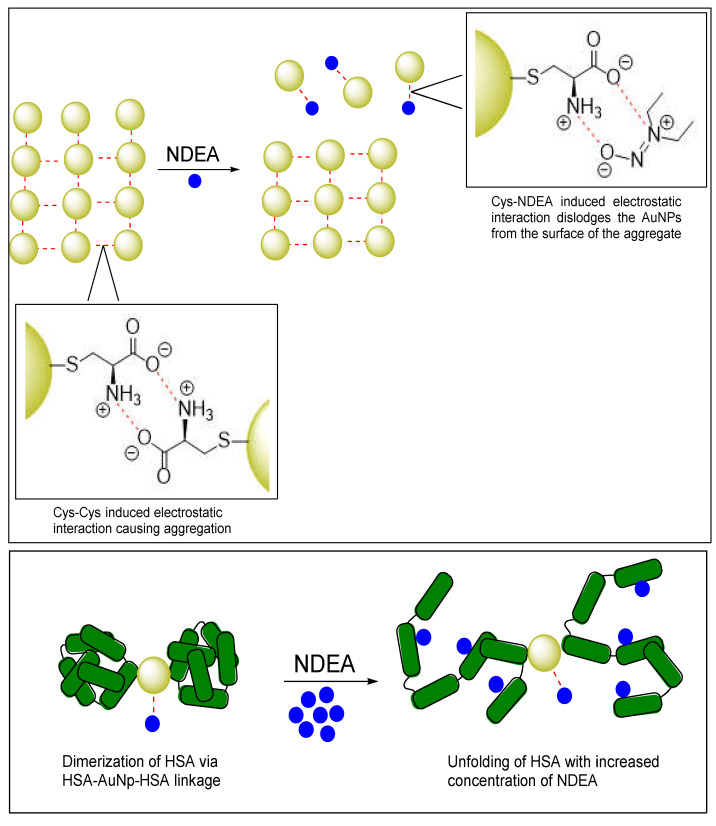
Schematic illustration of the proposed interaction mechanism NDEA, Cys-AuNPs, and HSA. *Top panel:* Cys-AuNPs (gold spheres) aggregate through electrostatic interactions with NDEA molecules (blue dots), as indicated by red dashed lines. *Bottom panel:* HSA protein (green rods) dimer is shown in its unfolded state (**left**) and after conformational changes induced by NDEA titration (**right**).

**Figure 7 sensors-25-05505-f007:**
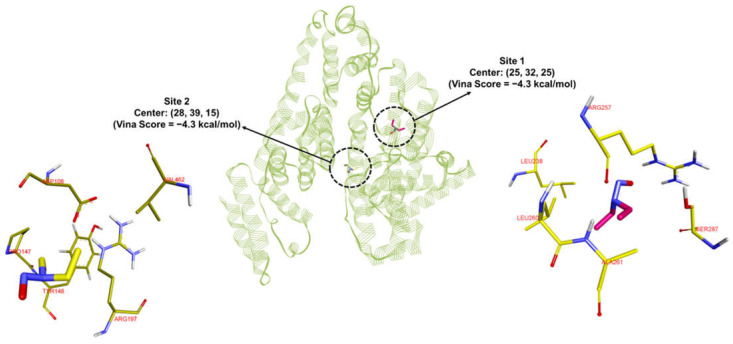
The top two binding sites were identified by the CB-Dock2 web server. For site 1, the carbon atoms of NDEA are represented in magenta, nitrogen in blue, and oxygen in red in the stick model. The same scheme was applied to the binding site residues, except that their carbon atoms are shown in yellow, while nitrogen and oxygen remain in blue and red, respectively. For site 2, the carbon atoms of NDEA are shown in yellow, with nitrogen in blue and oxygen in red. Similarly, the binding site residues follow this scheme, with their carbon atoms depicted in dark yellow.

**Figure 8 sensors-25-05505-f008:**
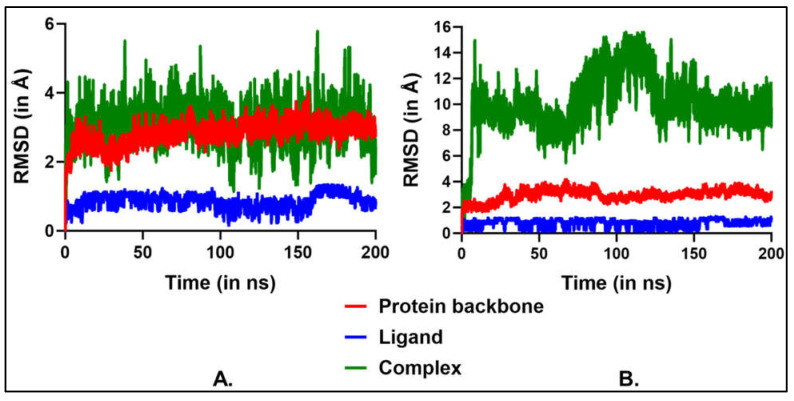
RMSD plots from the MD simulations for (**A**) Site 1 and (**B**) Site 2, as predicted by CB-Dock2 using AutoDock Vina.

**Table 1 sensors-25-05505-t001:** MM-GBSA-derived binding energies for the protein–ligand complexes at each binding site, presented in kcal/mol.

Binding Sites	van der Waals (kcal/mol)	Electrostatics (kcal/mol)	Polar Solvation Energy (kcal/mol)	Non-Polar Solvation Energy (kcal/mol)	TOTAL ΔG (kcal/mol)
Site 1	−17.3	−0.5	6.1	−2.7	−14.4
Site 2	−15.6	−0.3	6.2	−2.5	−12.2

## Data Availability

The raw data supporting the conclusions of this article will be made available by the authors on request.

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
