# Peer review of "Colorimetric Detection of Nitrosamines in Human Serum Albumin Using Cysteine-Capped Gold Nanoparticles"

_sensors, 2025, doi:10.3390/s25175505_

Round 1
Reviewer 1 Report
Comments and Suggestions for Authors
The work introduces a novel application of cysteine-capped gold nanoparticles (CysAuNPs) for colorimetric detection of N-nitroso diethylamine (NDEA) in human serum albumin (HSA), combining experimental synthesis/characterization with computational modeling (molecular docking and dynamics simulations). This aligns well with current interests in rapid, low-cost sensors for genotoxic impurities, especially given recent FDA/EMA recalls of nitrosamine-contaminated drugs. However, the manuscript has notable weaknesses in structure, logical flow, data presentation, and language that could undermine its credibility. It requires major revisions to enhance clarity, scientific rigor, and readability.
- The authors should clearly emphasize in both the abstract and introduction how this work goes beyond the current state-of-the-art (e.g., what makes CysAuNPs in HSA so advantageous over alternatives?).
- The manuscript discusses serum albumin but does not extend to real biological samples (e.g., spiked human serum or pharmaceutical matrices). The practicality for clinical or industrial application needs to be more directly addressed with at least pilot tests or a critical discussion.
- The abstract is dense and too long. It should be concise, highlighting the problem, methodology, key results, and significance.
- The discussion of regulations and nitrosamine concerns is rather lengthy and could be condensed.
- A more explicit statement about the current limitations of state-of-the-art colorimetric approaches for nitrosamines is needed.
- Clarify the rationale behind certain design choices (e.g., why specifically cysteine-capped AuNPs for this target in this matrix?).
- While the Job plot is helpful, the logic for the two linear regions in Figure 5B is not entirely clear. Further explanation and, if possible, a comparison with similar systems in literature would help.
- The molecular docking/MD simulation discussion is comprehensive, but the link between computational and experimental findings can be emphasized more clearly.
- Add more critical evaluation in Conclusion — what are the limitations (e.g., possible interferences, selectivity for other nitrosamines, effects of real sample matrices)?
- No selectivity studies against other nitrosamines or interferents.
- All experiments appear to use buffer or simplified systems. Demonstration in diluted serum, plasma, or a relevant pharmaceutical excipient matrix is vital for the field.
- Even just “spiking-and-recovery” results would be valuable.
- The mechanism (aggregation, electrostatics, unfolding) could be elucidated by visual evidence—e.g., TEM/UV-vis of sensor with/without NDEA in real-time, or more direct evidence for protein conformational change (CD spectroscopy, DLS results, etc.).
- Ensure all figures are of high enough resolution and fully self-explanatory with clear labels and legends.
-->The horizontal axis of Figure 1B is covered.
-->The scale units are missing in Figure 1C.
- The references list is extensive (83 entries), but ensure they are current and relevant.
Author Response
Reviewer 1’s Comment: English can be improved
Our Response: We have carefully edited the manuscript to address the minor grammatical errors in the revision.
Reviewer 1’s Comment: The work introduces a novel application of cysteine-capped gold nanoparticles (CysAuNPs) for colorimetric detection of N-nitroso diethylamine (NDEA) in human serum albumin (HSA), combining experimental synthesis/characterization with computational modeling (molecular docking and dynamics simulations). This aligns well with current interests in rapid, low-cost sensors for genotoxic impurities, especially given recent FDA/EMA recalls of nitrosamine-contaminated drugs. However, the manuscript has notable weaknesses in structure, logical flow, data presentation, and language that could undermine its credibility. It requires major revisions to enhance clarity, scientific rigor, and readability.
Our Response: We are grateful to Reviewer 1 for the positive comment on our manuscript. We have revised the manuscript to address the reviewer’s comments and concerns in the revised manuscript.
Reviewer 1’s Comment: The authors should clearly emphasize in both the abstract and introduction how this work goes beyond the current state-of-the-art (e.g., what makes CysAuNPs in HSA so advantageous over alternatives?).
Our Response: We agree with the reviewer on this comment. The combined use of molecular dynamic simulation and colorimetric experiment provided complementary data that allows robust analysis of NDEA in serum samples. In addition, the low cost of the UV-visible spectrophotometer, easy preparation, and optical sensitivity of CysAuNPs sensors are desirable, allowing low detection limit of the CysAuNPs sensors capable of detecting as low as 0.35 µM NDEA in serum albumin samples, making the protocol an attractive sensor for rapid detection of nitrosamines in biological samples.
Reviewer 1’s Comment: The manuscript discusses serum albumin but does not extend to real biological samples (e.g., spiked human serum or pharmaceutical matrices). The practicality for clinical or industrial application needs to be more directly addressed with at least pilot tests or a critical discussion.
Our Response: This is a welcome comment. We recognized this need for real sample analysis. This is ongoing research that will be communicated in the future manuscript. However, as a pioneer study, the colorimetric data and molecular dynamic simulation data are robust and compelling. Future studies include using CysAuNPs to detect nitrosamines in real biological samples including human serum, urine samples, environmental, and pharmaceutical samples.
Reviewer 1’s Comment: The abstract is dense and too long. It should be concise, highlighting the problem, methodology, key results, and significance.
Our Response: We are grateful to the reviewer for this comment. However, considering multiple analytical methods employed in this study (CysAuNPs synthesis, colorimetric and molecular dynamics simulation), we belief the length of this abstract (283 words) is adequate.
Reviewer 1’s Comment: The discussion of regulations and nitrosamine concerns is rather lengthy and could be condensed.
Our Response: We have revised and condensed this section as much as possible without losing the content in the revision.
Reviewer 1’s Comment: A more explicit statement about the current limitations of state-of-the-art colorimetric approaches for nitrosamines is needed.
Our Response: This is a valuable comment, and we agree with the reviewer. There are limitations including investigating more nitrosamines, potential interference, and selectivity for other nitrosamines or other structurally related compounds, as well as an analysis of NDEA in real samples. Nonetheless, the result of the combined use of CysAuNPs sensors for colorimetric analysis and molecular dynamic simulation study for NDEA detection in serum albumin is promising and provides preliminary results that lay the foundation for future studies. We have included these limitations and statement in the conclusion section of the revised manuscript.
Reviewer 1’s Comment: Clarify the rationale behind certain design choices (e.g., why specifically cysteine-capped AuNPs for this target in this matrix?).
Our Response: The is a welcome comment. CysAuNPs sensors were selected for NDEA detection in this study because of the attractive optical properties of CysAuNPs that allow sensitive detection of the analyte at low concentrations and the stability of CysAuNPs in solution. Human serum albumin was selected because of its physiological role in humans. Human serum albumin is the most abundant serum protein in the cardiovascular system, and a transporter of drugs and metabolites to various targets in the human body, at physiological conditions. As a drug and a metabolite transporter in the human body, serum aluminum can potentially bind with NDEA, with deleterious health implications for humans. We have included this statement in the revised manuscript.
Reviewer 1’s Comment: While the Job plot is helpful, the logic for the two linear regions in Figure 5B is not entirely clear. Further explanation and, if possible, a comparison with similar systems in literature would help.
Our Response: We appreciate the Reviewer’s comment very much. The result of our Job plot compared favorably with the results of molecular dynamic simulations (which we have included in the revised manuscript). We presented the two linear regions that coincide with the result of the Job plot. To our knowledge, this is the first study on CysAuNPs sensors for colorimetric NDEA detection in serum albumin, and we do not have similar prior literature for comparison at this time.
Reviewer 1’s Comment: The molecular docking/MD simulation discussion is comprehensive, but the link between computational and experimental findings can be emphasized more clearly.
Our Response: We are very grateful to the reviewer for this comment. We have included a discussion linking experimental work with the results of MD simulation study. The Job plot indicating a 2:1 NDEA:HSA stoichiometry provides experimental evidence that more than one NDEA molecule can associate with the HSA protein. This observation aligns with our proposed binding mechanism and molecular docking and MD simulation results, which revealed that NDEA can bind at two distinct sites on HSA with comparable binding free energies. These findings suggest that HSA can simultaneously accommodate multiple NDEA molecules at different binding regions, which explains the 1:2 stoichiometry detected experimentally. Thus, the congruence of the Job plot data with the computational binding site predictions validates the mechanistic interpretation of NDEA–HSA interactions and their role in driving the optical response of the CysAuNP-based sensor. We have included this statement in the revised manuscript.
Reviewer 1’s Comment: Add more critical evaluation in Conclusion — what are the limitations (e.g., possible interferences, selectivity for other nitrosamines, effects of real sample matrices)?
Our Response: This is an excellent suggestion. There possible limitations that will be evaluated and reported in future studies including potential interferences, and selectivity for other nitrosamines or other structurally related compounds.In addition, an evaluation of sensor performance on detection of NDEA in real samples will be carried out. Nonetheless, the result of the combined use of CysAuNPs sensors for colorimetric analysis and molecular dynamic simulation study for NDEA detection in serum albumin yielded interesting results that lay the foundation for future studies. We have described these limitations in the conclusion section of the revised manuscript.
Reviewer 1’s Comment: No selectivity studies against other nitrosamines or interferents.
Our Response: We have noted this valuable comment. As described in our previous response above, we will conduct selectivity studies against other nitrosamines or interferents in future studies.
Reviewer 1’s Comment: All experiments appear to use buffer or simplified systems. Demonstration in diluted serum, plasma, or a relevant pharmaceutical excipient matrix is vital for the field.
Our Response: This is a legitimate comment. However, the experiments were conducted in serum albumin samples at physiological conditions that mimic real-world plasma sample environment. . This is relevant since serum albumin is the most abundant protein in blood plasma. As a pioneer study, the colorimetric data and molecular dynamic simulation data are robust and compelling. Future studies will include using CysAuNPs to detect more nitrosamines in real biological samples including human serum, urine samples, environmental, and pharmaceutical samples.
Reviewer 1’s Comment: Even just “spiking-and-recovery” results would be valuable.
Our Response: We thank the reviewer for this valuable comment. However, this is an ongoing, extensive research study, and the results, including those of spiking and recovery, will be communicated in future manuscripts
Reviewer 1’s Comment: The mechanism (aggregation, electrostatics, unfolding) could be elucidated by visual evidence—e.g., TEM/UV-vis of sensor with/without NDEA in real-time, or more direct evidence for protein conformational change (CD spectroscopy, DLS results, etc.).
Our Response: We appreciate the reviewer’s comment. The interaction of NDEA with HSA has minimal impact on the HSA UV-visible absorption. However, the addition of NDEA and CysAuNPs into serum albumin resulted in a white cloudy solution, which is more conspicuous at higher NDEA concentrations, suggesting serum albumin protein agglomeration. We have included this statement in the revised manuscript.
Reviewer 1’s Comment: Ensure all figures are of high enough resolution and fully self-explanatory with clear labels and legends. -->The horizontal axis of Figure 1B is covered. -->The scale units are missing in Figure 1C.
Our Response: We concur with the reviewer’s comments on the quality of some figures, particularly Figure 1 and Figure 2. We have provided high-quality Figure 1 and Figure 2 in the revision. We have included the unit in Figure 1C. In addition, we have provided a better Figure 3 to correct wrong auto-correct HAS in the revision. We have included the unit for each parameter in Table 1. We have also spelled out the variable for clarity.
Reviewer 1’s Comment: The references list is extensive (83 entries), but ensure they are current and relevant.
Our Response: All the cited references are relevant to this study.
Reviewer 2 Report
Comments and Suggestions for Authors
The article “Colorimetric Detection of Nitrosamines in Human Serum Albumin Using Cysteine-Capped Gold Nanoparticles” describes a new method for detecting nitrosamines in human serum albumin using gold nanoparticles. The nanoparticles are characterized by various methods, including UV-Vis spectroscopy, fluorescence, dynamic light scattering (DLS), and transmission electron microscopy (TEM). In the presence of N-nitrosodiethylamine, an increase in absorbance at approximately 378 nm is observed in the CysAuNPs-HSA system. The method has a relatively low detection limit. The authors also conducted molecular docking and molecular dynamics studies. Despite the good quality of the work, there are significant areas that need improvement, as described below:
1. The authors do not demonstrate the selectivity of the proposed method in any way. Without this, the proposed method cannot be considered a reliable detection method. What spectral response will be observed, for example, with the addition of amines?
2. How do various interfering molecules and ions affect the detection of N-nitrosodiethylamine?
3. Why was a fluorimetric study of the HSA-CysAuNPs-NDEA system not performed? Perhaps using fluorimetry could increase the sensitivity of the method.
4. From the UV-Vis data, please determine the binding constant to HSA in solution.
5. What changes in the UV-Vis spectra are observed with a large addition of NDEA? It is possible that the linear range of NDEA concentrations to be determined is greater than 9 µM.
6. What is the reason for the decrease in absorbance at a concentration of 5 µM NDEA?
7. There are not enough additional data points and error bars in Figure 5B. The R² value is very low for the first calibration line.
8. Compare the proposed method with other methods for detecting nitrosamines (considering detection limit, cost, labor intensity, etc.).
Author Response
Reviewer 2’s Comment: The article “Colorimetric Detection of Nitrosamines in Human Serum Albumin Using CysteineCapped Gold Nanoparticles” describes a new method for detecting nitrosamines in human serum albumin using gold nanoparticles. The nanoparticles are characterized by various methods, including UV-Vis spectroscopy, fluorescence, dynamic light scaÄ´ering (DLS), and transmission electron microscopy (TEM). In the presence of N-nitrosodiethylamine, an increase in absorbance at approximately 378 nm is observed in the CysAuNPs-HSA system. The method has a relatively low detection limit. The authors also conducted molecular docking and molecular dynamics studies. Despite the good quality of the work, there are significant areas that need improvement, as described below:
Our Response: We are grateful to Reviewer 2 for the positive comment on our manuscript. We have revised the manuscript to address the Reviewer’s comments and concerns in the revised manuscript.
Reviewer 2’s Comment: The authors do not demonstrate the selectivity of the proposed method in any way. Without this, the proposed method cannot be considered a reliable detection method. What spectral response will be observed, for example, with the addition of amines? How do various interfering molecules and ions affect the detection of N-nitrosodiethylamine?
Our Response: We appreciate the Reviewer’s comments. Please see our response to Reviewer 1 in a related comment. We recognized this need for real sample analysis, selectivity, and interference studies. These are ongoing research studies that will be communicated in future manuscripts. However, as a pioneer study, the colorimetric and molecular dynamic simulation data are robust and compelling. Future studies include using CysAuNPs to detect more nitrosamines in real biological samples, including human serum, urine samples, environmental, and pharmaceutical samples. Future study also includes the determination of the binding affinity and the use of circular dichroism spectroscopy to probe the molecular interaction of NDEA binding with serum albumin, which may provide additional insight into the binding mechanism and structural and molecular changes of serum albumin because of interaction with NDEA. These are ongoing research and will be communicated in future manuscripts.
Reviewer 2’s Comment: Why was a fluorimetric study of the HSA-CysAuNPs-NDEA system not performed? Perhaps using fluorimetry could increase the sensitivity of the method.
Our Response: We appreciate the reviewer’s comment on the possible use of a fluorometric study of the HSA-CysAuNPs-NDEA system. First, serum albumin and CysAuNPs have excellent UV-visible absorption properties that allowed low detection as low as 0.35 µM NDEA in serum albumin in this study. A fluorescence study will lower the detection limit and provide better selectivity. However, UV-visible spectrometers are commonly available in most laboratories and are relatively cheaper than a spectrofluorometer, allowing low-cost analysis. Nonetheless, we will explore the possible use of fluorescence spectroscopy for sensitive detection of nitrosamines in serum albumin in a future study.
Reviewer 2’s Comment: From the UV-Vis data, please determine the binding constant to HSA in solution.
Our Response: We appreciate the reviewer’s comments. There are many prospects and future studies, including the determination of selectivity, binding constant, and thermodynamics of the interaction of NDEA with serum albumin. Those studies are beyond the scope of this pioneer study . We will explore those studies in future manuscripts.
Reviewer 2’s Comment: What changes in the UV-Vis spectra are observed with a large addition of NDEA? It is possible that the linear range of NDEA concentrations to be determined is greater than 9 µM.
Our Response: This is a great comment. Changes in UV-visible spectra are NDEA concentration- dependent, as shown in the graph we presented. Indeed, the linear range of NDEA concentrations may be larger than 9 µM, which will be incredible. This study focused on testing the ability of the sensor to detect low amounts of NDEA in environmental and biomedically relevant concentrations..
Reviewer 2’s Comment: What is the reason for the decrease in absorbance at a concentration of 5 µM NDEA?
Our Response: A decrease in absorbance at 5 µM NDEA concentration is likely due to changes in dynamics and equilibrium due to NDEA interaction with serum albumin. We have included this statement in the text of the revised manuscript. Interestingly, this is the NDEA concentration, where the Job plot indicated a 1:2 mole ratio of NDEA to HSA in NDEA-CysAuNPs-HSA.
Reviewer 2’s Comment: There are not enough additional data points and error bars in Figure 5B. The R² value is very low for the first calibration line.
Our Response: This is a great comment. Several factors, including instrument sensitivity, sample absorbance and the volume of analyte solution that can be reliably dispensed determined the suitable NDEA concentration range for this study. Using a syringe in this study, dispensing NDEA samples lower than 2.46 µM NDEA is unreliable. The correlation coefficient for the first calibration curve of 0.8643 is lower than that of the second calibration curve (R2 =0.9938). However, R2 of 0.8643 is acceptable for this type of colorimetric study at 2.46 µM -9 2 µM NDEA range.
Reviewer 2’s Comment: Compare the proposed method with other methods for detecting nitrosamines (considering detection limit, cost, labor intensity, etc.).
Our Response: These are valuable comments and suggestions by the reviewer. We have compared the advantages of our techniques with the contemporary methods of nitrosamine analysis, including simple sample preparation, direct measurement, low-cost, and rapidity in the revised manuscript. In addition, we have compared the observed detection limit from this study with some reported detection limits in the literature. Our observed LOD is comparable with the reported limit of detection of 0.66pm colorimetric detection of N-nitrosamines in aqueous solutions and a limit of detection (LOD) ranging between 0.06 and 0.5 µM for chemiluminescence detection of N-nitrosamines in water samples. We have included this statement in the revision.
Reviewer 3 Report
Comments and Suggestions for Authors
The article entitle “Colorimetric Detection of Nitrosamines in Human Serum Albumin Using Cysteine Capped Gold Nanoparticles” is an interesting work that describing the obtention of cysteine-gold nanoparticles (CysAuNPs) for colorimetric detection of N-nitroso diethylamine (NDEA). The article is well structured, yet some changes should be done to improve the manuscript:
I think that more specific references should be provided about N-nitroso diethylamine (NDEA). The introduction is very general about nitrosamines.
In the Materials and Methods section, the authors should present the experiments performed using the UV-Visible technique separately. This information is confusing.
Figure 3C shows increases in absorbance with increasing NDEA content. Is this relationship linear at 378 nm? The authors should present the data in tables.
Another aspect the authors should review is Figure 5B. Why do they refer to two calibration curves? The authors should clarify; for me, it's just one.
Reviewing the data in Figure 5B, the most dilute solutions should be treated by statistical methods and eliminated if necessary.
I believe that the data in Figure 5B should be remeasured or include higher concentration NDEA data.
What calibration curve was used to determine the LOD? Could you determine the limit of quantification?
In the Job plot of the absorbance versus mole fraction, more data is required
Author Response
Reviewer 3’s Comment: The article entitle “Colorimetric Detection of Nitrosamines in Human Serum Albumin Using Cysteine Capped Gold Nanoparticles” is an interesting work that describing the obtention of cysteine-gold nanoparticles (CysAuNPs) for colorimetric detection of N-nitroso diethylamine (NDEA). The article is well structured, yet some changes should be done to improve the manuscript:
Our Response: We are grateful to Reviewer 3 for the time and effort to review our manuscript. We are also thankful to Reviewer #3 for the positive comment on our manuscript. We have revised the manuscript to address the reviewer’s comments and concerns in the revised manuscript.
Reviewer 3’s Comment: I think that more specific references should be provided about N-nitroso diethylamine (NDEA). The introduction is very general about nitrosamines.
Our Response: Nitrosamines including NDEA are all toxic contaminants that informed our general discussion of nitrosamines. We provided several NDEA discussions and citations including its detections, in the revised manuscript.
Reviewer 3’s Comment: In the Materials and Methods section, the authors should present the experiments performed using the UV-Visible technique separately. This information is confusing.
Our Response: We agree with the reviewer. We have reorganized the material and methods section for clarity to address this comment in the revised manuscript.
Reviewer 3’s Comment: Figure 3C shows increases in absorbance with increasing NDEA content. Is this relationship linear at 378 nm?
Our Response: The relationship between the absorbance and NDEA concentration at 378 nm was a double curve calibration as previously discussed in a related comment.
Reviewer 3’s Comment: The authors should present the data in tables. Another aspect the authors should review is Figure 5B. Why do they refer to two calibration curves? The authors should clarify; for me, it's just one.
Our Response: We value the reviewer’s comment. We have provided a detailed discussion of Figure 5B in the revised paper. Figure 5B calibration curve has two distinct regions, known as a double calibration curve. A double curve is indicative that a single linear model cannot represent the data due to multiple reasons, including changes in instrumental sensitivity and response to NDEA concentration. A similar double calibration curve has been reported in the literature (Liu Boshi , Huang Renliang , Yu Yanjun , Su Rongxin , Qi Wei , He Zhimin. Gold Nanoparticle-Aptamer-Based LSPR Sensing of Ochratoxin A at a Widened Detection Range by Double Calibration Curve Method. Frontiers in Chemistry, 2018, 6, 1-9. DOI=10.3389/fchem.2018.00094). We have included this discussion and provided relevant citations in the revised manuscript
Reviewer 3’s Comment: Reviewing the data in Figure 5B, the most dilute solutions should be treated by statistical methods and eliminated if necessary.
Our Response: We appreciate this comment. The reviewer is correct about sample preparation challenges at low NDEA analyte concentration. However, after rigorous and careful sample preparation optimization, we reliably prepared the NDEA concentrations at this level.
Reviewer 3’s Comment: I believe that the data in Figure 5B should be remeasured or include higher concentration NDEA data.
Our Response: We appreciate this comment. However, several factors, including instrument sensitivity, CysAuNPs optical properties, and serum albumin UV-Visible absorption, determined the suitable NDEA concentration range used in this study. The optimum working range for this is study is between 2.46 µM and 8.61 µM NDEA
Reviewer 3’s Comment: What calibration curve was used to determine the LOD? Could you determine the limit of quantification? In the Job plot of the absorbance versus mole fraction, more data is required
Our Response: This is a valuable comment, which we appreciate. We have clarified this comment in the revision. Also, we have calculated the limit of quantification as suggested by the reviewer and included this in the revised manuscript.
Depending on the NDEA concentrations, two possible calibration curves can be constructed for the determination of unknown NDEA samples. Using the first calibration curve, the obtained figures-of-merit highlighted by a high correlation, the calculated low limit of detection (LOD) of 0.210 µM and the limit of quantitation (LOQ) of 0.70 µM show the sensitivity of CysAuNPs for NDEA detection in serum albumin. The LOD was calculated as the where s is the standard deviation of blank triplicate analyses and m is the slope of the calibration curve regression equation. The LOQ was calculated as the where s is the standard deviation of blank triplicate analyses and m is the slope of the calibration curve regression equation. The use of the second calibration curve resulted in LOD of 0.49 µM and LOQ of 1.62 µM for NDEA detection in serum albumin. Either calibration curve can be used for NDEA detections depending on the NDEA concentration range. Our observed LOD is comparable with the reported limit of detection of 0.66pm colorimetric detection of N-nitrosamines in aqueous solutions [48] and a limit of detection LOD ranging between 0.06 and 0.5 µM for chemiluminescence detection of N-nitrosamines in water samples [49]. A fluorescence study will lower the detection limit and provide better selectivity. We have included this statement in the revised manuscript.
Regarding more data on the Job plot, the point to inflection is well defined. Adding more data points will not change the 2:1 ratio.
Round 2
Reviewer 1 Report
Comments and Suggestions for Authors
The authors had endeavored to address the relevant issues. The manuscript with its current status can be accepted for publication.
Author Response
Reviewer 1 Comment: The authors had endeavored to address the relevant issues. The manuscript with its current status can be accepted for publication.
Our Response: We are grateful to Reviewer 1 for the recommendation of accepting of our manuscript in its current form for publication in Sensors. We appreciate the reviewer 1’s invaluable and positive feedback that has strengthened the quality of our manuscript.
Reviewer 2 Report
Comments and Suggestions for Authors
The authors did not provide evidence of the selectivity of the proposed method, which is extremely important for the sensor. The colorimetric and molecular dynamics simulation data do not demonstrate selectivity. Additionally, the calibration curves are plotted at only three points, which is unreliable and lacks reproducibility. It is recommended that the authors provide evidence of both selectivity and reproducibility.
Author Response
Reviewer 2’s Comment: The authors did not provide evidence of the selectivity of the proposed method, which is extremely important for the sensor. The colorimetric and molecular dynamics simulation data do not demonstrate selectivity. Additionally, the calibration curves are plotted at only three points, which is unreliable and lacks reproducibility. It is recommended that the authors provide evidence of both selectivity and reproducibility.
Our response. We have responded to this comment in the first manuscript revision. As we previously stated, we recognized this need for real sample analysis, selectivity, and interference studies. These are ongoing research studies that will be communicated in future manuscripts. Reviewer 1 and Reviewer 3 raised the same selectivity comments, and both were satisfied with our response. In fact, Reviewer 1 and Reviewer 3 have recommended the acceptance of our manuscript in its current form for publication in Sensors. Nonetheless, we are working toward the selectivity and interference studies. These are ongoing research and will be communicated in future manuscripts.
Reviewer 3 Report
Comments and Suggestions for Authors
In this new revision, the authors have satisfactorily responded to the criticism in my previous report. As a result, I recommend the acceptance for publication in the current state.
Author Response
Reviewer 3’s Comment: In this new revision, the authors have satisfactorily responded to the criticism in my previous report. As a result, I recommend acceptance for publication in the current state.
Our Response: We are grateful to Reviewer 3 for the recommendation to accept our manuscript in its current form for publication in Sensors. We appreciate reviewer 3’s invaluable and positive feedback that has strengthened the quality of our manuscript.